# OpenReview forum: "VAFL: Vector-field Assisted Functional Layer for Equal-Compute Multi-modal Learning"
_ICLR.cc/2026/Conference — ICLR 2026 Conference Withdrawn Submission_

### Official Review · Reviewer_pPkn · 2025-10-16

**Soundness:** 1
**Presentation:** 1
**Contribution:** 1
**Rating:** 0
**Confidence:** 4

**Summary:**

The paper introduces a functional layer that computes the energy of the latent features. The layer is in charge of energy descent both in training and testing. The trained model achieves marginal improvement compared to the baseline.

**Strengths:**

1. The experiment details are given

**Weaknesses:**

1. Poor presentation. The authors do not provide enough background for the paper, such as an introduction to the energy-based model, the motivation for the energy definition, and related papers. There is no reference to existing works in the Related Works section.
2. Lack of testing on the downstream performance. While the trained model achieves certain improvements in the pretraining metrics, there is no test on its downstream performance. Evidence suggests that pretraining performance might not correlate with the downstream one [1].
3. Wrong usage of Langevin dynamics. The authors mention that the noise term is ignored during inference. Without the noise term, the ODE cannot able to be turned into a probability flow that transports between probability distributions, violating the EBM formulation. This practice suggests that the performance improvement might not come from the EBM formulation but from artifacts, e.g., more FLOPS.
4. The figures are not informative. For example, Figure 3 is repetitive given the result in Table 1.
5. The proposed metric SOMA is not novel and confusing. The Distinct-2 exists in [2]. The quality measurement is just a kind of average over the losses. The Stability metric is confusing, as it achieves the optimal when all the latents are not changed. This violates the motivation of refining the latents.
6. The vision dataset is not convincing. Could this method generalize to larger-scale image datasets like ImageNet1K?

[1] Denoising diffusion autoencoders are unified self-supervised learners. ICCV 2023.

[2] A diversity-promoting objective function for neural conversation models. NAACL 2016.

Overall, I think this paper reads like an LLM auto-generated paper, and it is far below ICLR's acceptance criteria.

**Questions:**

Could the author formalize the definition of energy in the EBM formulation? For example, what is the target distribution that you are trying to model, and what is the intuition behind the energy definition?

---

### Official Review · Reviewer_6twf · 2025-10-22

**Soundness:** 2
**Presentation:** 1
**Contribution:** 1
**Rating:** 2
**Confidence:** 4

**Summary:**

The paper presents VAFL. a method that applies two steps of Langevin dynamic to refine multi-modal representations under equal-compute constraints. It also introduces SOMA as evaluation metric that linearly combines quality, diversity and stability.

**Strengths:**

- The idea of refining hidden representations rather than scaling parameters is conceptually interesting

**Weaknesses:**

- The proposed method is essentially a minor post-hoc gradient update with a learned MLP. Similar approaches have been explored extensively, such as TENT [1] o PPLM [2]. The work does not have any comparison with such post-training refinement methods and is remained unclear what is the advantage of proposed method.
-  The experiments are extremely shallow. It shows improvements of 7-9% on small scale datasets that are within noise margins. There is no comparison to any real baselines.
- The introduced SOMA is poorly motivated and it appears that it is designed to inflate reported gains. The proposed weights in linear combination are chosen without any justification or sensitivity analysis.

[1] Wang, Dequan, Evan Shelhamer, Shaoteng Liu, Bruno Olshausen, and Trevor Darrell. "Tent: Fully test-time adaptation by entropy minimization." arXiv preprint arXiv:2006.10726 (2020).

[2] Dathathri, Sumanth, Andrea Madotto, Janice Lan, Jane Hung, Eric Frank, Piero Molino, Jason Yosinski, and Rosanne Liu. "Plug and play language models: A simple approach to controlled text generation." arXiv preprint arXiv:1912.02164 (2019).

**Questions:**

N/A

---

### Official Review · Reviewer_qGux · 2025-10-28

**Soundness:** 1
**Presentation:** 2
**Contribution:** 1
**Rating:** 0
**Confidence:** 5

**Summary:**

This paper proposes VAFL, an energy-based refinement mechanism applying Langevin dynamics steps to multi-modal transformer representations. Evaluated on WikiText-2, CIFAR-10, and Speech Commands, it shows perplexity/MSE improvements with minimal FLOPs overhead. A new SOMA metric is introduced to jointly measure quality, diversity, and stability.

**Strengths:**

-  I am genuinely struggling to find any.

**Weaknesses:**

- The exposition is unclear, the paper does not specify how the representations are combined/unified, and then unembedded for the perplexity/MSE evaluations.
- No baselines or comparison are provided, either with other multi-modal methods, refinement approaches, or simply training the base model longer with equivalent compute. Improvements are uninterpretable without context.
- Limited scope: Small model (31M params), short training (10k steps), standard benchmarks, training objective-only evaluation. No downstream tasks or realistic scale experiments.
- SOMA metric unjustified: Arbitrary weights and components with no principled design or external validation. Only used to evaluate the authors' own method.
- Weak motivation: No explanation for why Langevin refinement helps multi-modal learning specifically. Missing theoretical analysis or intuition.
- Insufficient related work: Entirely omits discussion of modern multi-modal methods and other related work.

Overall: A simple idea with preliminary validation; this reads as an incomplete course project rather than a mature research contribution, well below the standards of rigor expected at ICLR.

**Questions:**

See weaknesses

---

### Official Review · Reviewer_yJ67 · 2025-10-30

**Soundness:** 1
**Presentation:** 1
**Contribution:** 1
**Rating:** 0
**Confidence:** 4

**Summary:**

The paper proposes VAFL (Vector-field Assisted Functional Layer), an energy-based refinement module that applies short Langevin-style updates (K=2) on hidden states from a Transformer to improve text/image/audio under a self-imposed “equal-compute” budget.

**Strengths:**

1. Addresses compute-efficiency as a practical constraint.
2. Implementation is simple (a small per-position MLP “energy” and 1–2 refinement steps) with modest overhead

**Weaknesses:**

1. Misleading framing: The method is presented as an “energy-based Langevin refinement,” but with τ=0 it degenerates into a deterministic residual update. This is conceptually equivalent to a cheap adapter block, not a genuine energy-based method.
2. Toy-scale evaluation: Experiments are confined to WikiText-2, CIFAR-10, and Speech Commands with ~30M-param models. These are far below community standards and provide no evidence that the method works in realistic multi-modal or large-scale settings.
3. Lack of fair baselines: There is no comparison to equal computation alternatives, such as LoRA, adapters, or simply adding one extra Transformer block under the same compute budget. This makes the reported gains unconvincing.

**Questions:**

1. EBM vs. deterministic refiner: With τ=0, what is gained by the EBM/Langevin framing over a standard residual adapter/refiner?

---

### Official Review · Reviewer_WhWL · 2025-11-02

**Soundness:** 1
**Presentation:** 1
**Contribution:** 2
**Rating:** 0
**Confidence:** 4

**Summary:**

This paper proposes VAFL (Vector-field Assisted Functional Layer), a lightweight energy-based refinement layer that improves multi-modal model performance under strict compute constraints. It refines hidden representations through short Langevin dynamics guided by a learned energy function, without modifying the base architecture. The paper also introduces a new SOMA (Synergistic Optimization for Multi-modal Assessment) metric to jointly evaluate quality, diversity, and stability across text, image, and audio modalities.

**Strengths:**

1. New idea for improving compute-efficiency: the paper leverages energy-based refinement but not the conventional way of modifying architecture components, which is somewhat novel.
2. Empirical gains under strict equal-compute constraint: The paper shows some improvements across text, vision, and audio tasks with only small amount of extra FLOPs.

**Weaknesses:**

This paper evidently does not meet the standard of a clear, well-structured, and comprehensive conference style paper. There's limited discussion on related work. The format and experiment depth looks more like a class project rather than an academic paper. The method section lacks clarity. A lot of parts (e.g., section 4.6) are entirely missing.

**Questions:**

NA

---

### Author Response · Authors · 2025-11-14
**clarifying scope and planned revisions**

Dear reviewers, AC, and SAC,

Thank you for your careful reading and constructive comments. I agree that the current version of the paper is not yet conceptually clean or sufficiently precise, and the responsibility for this is entirely mine.

Based on your feedback and my own re-reading, I now see clear issues in
(i) how the EBM/DSM framing is presented
(ii) how the latent refinement view is justified
(iii) how equal-compute and extra parameters are specified and evaluated.
The current draft also does not clearly separate what is standard theory, what is empirical observation, and what is my own intuition about representation geometry.

For the revision, I am planning a substantial and focused rewrite rather than minor editing. In particular, I will:
1. Reframe VAFL explicitly as a practical latent refinement layer that borrows DSM/Langevin tools, not as a new EBM theory.
2. Clarify in Sections 1–3 which parts follow existing EBM/DSM results and which parts are heuristic engineering choices applied in latent space.
3. Make the experimental protocol more standard and reproducible: explicit training/inference FLOPs definitions, equal-compute tolerances, and additional parameter counts reported alongside all results.
4. Base all main claims only on standard metrics (PPL, CE, accuracy, MSE); more ad-hoc measures and scaling observations will be moved to the appendix as optional analysis.
5. Add analyses that directly address your concerns, including energy–loss correlation, behaviour on easy vs. hard samples, and explicit failure cases.

My aim is to narrow the claims, strengthen the logical structure, and make the method and experiments easier to reproduce and critique. Regardless of the final decision, I appreciate the detailed feedback and will use it to make this line of work more rigorous.

Thank you again for your time and evaluation.

---

### Note · Authors · 2025-12-03

**Comment:**

After submission, I conducted multi-seed experiments across various scales in text, image, audio, and multimodal domains to ensure fair comparison under equal-compute constraints. Despite my best efforts to perform thorough validation within limited GPU resources, the proposed method did not show statistically significant improvements due to high variance across seeds in small-scale experiments. In particular, I was unable to reproduce consistent improvements over baselines in the image and audio domains. While I observed meaningful improvements (~25% PPL reduction) in large-scale frozen backbone experiments, I lacked sufficient resources to conduct the ablation studies necessary to verify whether this effect stems from the proposed mechanism and whether it generalizes to other modalities. I have decided to withdraw as the current results do not sufficiently support the paper's core claim of "cross-modal generalization."
This was my first conference submission. I sincerely thank the reviewers for their valuable feedback, and the ICLR organizing committee and OpenReview platform for providing this opportunity. I will further develop the methodology based on this experience and return with sufficient experimental evidence.

**Withdrawal Confirmation:**

I have read and agree with the venue's withdrawal policy on behalf of myself and my co-authors.